# Evaluation of Five Plasma miRNAs as Biomarkers for Minimally Invasive Staging of Liver Fibrosis in β-Thalassaemia Patients

**DOI:** 10.3390/ijms26199543

**Published:** 2025-09-30

**Authors:** Sevgi Özkaramehmet, Savanna Andreou, Kristia Yiangou, Soteroula Christou, Michalis Hadjigavriel, Maria Sitarou, Katerina Pyrovolaki, Eleni Papanicolaou, Christina Flourou, Irene Savvidou, Panagiotis Boutsikos, Alexandra Mendoni, Marina Kleanthous, Marios Phylactides, Carsten W. Lederer

**Affiliations:** 1Molecular Genetics of Thalassaemia Department, The Cyprus Institute of Neurology & Genetics, 6 Iroon Avenue, Ayios Dometios, 2371 Nicosia, Cyprus; sevgio@cing.ac.cy (S.Ö.); savannaandreou@hotmail.com (S.A.); marinakl@cing.ac.cy (M.K.); 2Biostatistics Unit, The Cyprus Institute of Neurology & Genetics, 6 Iroon Avenue, Ayios Dometios, 2371 Nicosia, Cyprus; kristiay@cing.ac.cy; 3Thalassaemia Clinic Nicosia, Archbishop Makarios III Hospital, 1474 Nicosia, Cyprus; snchrthalcl@cytanet.com.cy (S.C.); savvidesiren@gmail.com (I.S.); p.boutsikos@yahoo.com (P.B.); alexmendoni1@gmail.com (A.M.); 4Thalassaemia Clinic Limassol, Limassol General Hospital, Kato Polemidia, 4131 Limassol, Cyprus; drmhad@gmail.com; 5Thalassaemia Clinic Larnaca, Larnaca General Hospital, 6301 Larnaca, Cyprus; msitarou@yahoo.gr (M.S.); k.pyrovolaki@shso.org.cy (K.P.); 6Internal Medicine Department, Nicosia General Hospital, Strovolos, 2029 Nicosia, Cyprus; elenipapanicolaou@gmail.com (E.P.); christinafl@hotmail.gr (C.F.)

**Keywords:** β-thalassaemia, liver fibrosis, microRNA, biomarker, fibrosis staging

## Abstract

Iron overload-driven liver fibrosis is a major concern in β-thalassaemia patients, but non-invasive or minimally invasive biomarkers for fibrosis staging remain limited. This study evaluated five plasma microRNAs (let-7a, miR-21, miR-29a, miR-34a, and miR-122) as potential markers for distinguishing liver fibrosis stages in β-thalassaemia. Plasma samples from 40 patients with fibrosis stages F0–F1 to F4 were analysed using RT-qPCR, normalised against the arithmetic mean of reference miRNAs miR-16 and miR-221. Expression levels of candidate miRNAs showed no statistically significant variation across stages, and logistic regression and ROC analyses revealed fair discriminatory performance for individual miRNAs and their combinations in selected stage comparisons. Notably, while for the discrimination of different fibrosis stages all five candidate miRNAs tested showed fair area-under-the-curve values between 0.7 and 0.8 individually and up to 0.917 in combination, none of these findings reached statistical significance. These results suggest that while the selected set of miRNAs reflects liver injury, its performance for precise fibrosis staging in β-thalassaemia is limited. A key cause for the low discriminatory power of these miRNAs may be the overall change of the blood miRNA transcriptome in haemoglobinopathies. The results indicate the need for validation in larger cohorts based on larger miRNA panels or the use of alternative source materials to improve diagnostic performance.

## 1. Introduction

Beta-thalassaemia is a hereditary haemoglobinopathy caused by reduced or absent β-globin expression. Affected individuals may be classified as asymptomatic carriers, non-transfusion-dependent thalassaemia (NTDT) patients, or transfusion-dependent thalassaemia (TDT) patients [1]. Clinical features of β-thalassaemia include variable degrees of anaemia, hepatosplenomegaly, and iron overload. NTDT patients suffer from iron overload due to increased intestinal iron absorption. By contrast, patients with TDT suffer from iron overload resulting from iron accumulation in their liver, primarily as a result of the frequent blood transfusions they receive as part of their treatment [1,2]. In addition to blood transfusions, the treatment approach for β-thalassaemia includes supportive therapy and complication-specific treatments, such as iron chelation therapy to combat the iron overload [3]. Absence or ineffectiveness of iron chelation therapy may exacerbate iron overload, and thus lead to significant organ dysfunction or failure, with the heart and liver being the organs most commonly affected [4].

In a healthy liver, there is a balance between the deposition and removal of extracellular matrix (ECM). Toxicity of iron deposits, however, results in a disruption of the normal hepatic regeneration and healing responses, in increased synthesis and decreased degradation of the ECM, and thus in the accumulation of excess ECM within the liver parenchyma [5]. This process leads to chronic injury and a build-up of scar tissue in the liver, i.e., liver fibrosis, as an initial hallmark of liver damage.

Early-stage liver fibrosis is reversible, but, if left untreated, it can lead to cirrhosis, organ dysfunction, and, eventually, hepatocellular carcinoma (HCC) [6,7]. Liver biopsy is considered the “gold standard” for detecting and staging liver fibrosis. However, due to its invasive nature, it is not used for routine patient screening. Instead, non-invasive biochemical and imaging techniques are used for regular evaluation of the liver [8], such as ultrasound-based elastography. The most established method is transient elastography (TE), which assesses liver stiffness as an indicative marker of fibrosis. Additionally, shear wave elastography (SWE) has been introduced, enabling real-time two-dimensional assessment of liver stiffness [9]. Both techniques are widely used in clinical practice; however, each has limitations. For instance, the accuracy of these techniques is affected by the age and body mass index of the patient, the width of the intercostal space, and the presence of ascites [9,10]. Additionally, TE may not be able to detect the early stages of fibrosis due to the one-dimensional nature of ultrasound elastography, which measures only a pre-specified area of the liver, increasing the risk of sampling error [11].

As an alternative to elastography, the potential of several plasma biomarkers for the diagnosis and staging of liver fibrosis is currently being investigated. Plasma biomarkers offer a cost-effective alternative to liver biopsy and imaging techniques, with minimal levels of invasiveness, sampling errors, and observer-related variability [12]. According to the literature, these biomarkers provide a high negative predictive value (NPV) but a poor positive predictive value (PPV) [12,13,14,15]. This makes them reliable for the detection of individuals without liver damage, but insufficient as stand-alone markers for the prediction of liver fibrosis, which thus still relies on validation by liver biopsy [13,15].

MicroRNAs (miRNAs) are increasingly used as diagnostic biomarkers and are readily accessible by isolation from plasma. They are small non-coding RNAs of approximately 22 nucleotides in length and act to regulate cellular and developmental pathways. As a common mode of action, they bind to the 3′-untranslated region of target mRNAs to achieve post-transcriptional regulation of gene expression as well as RNA silencing [16,17]. Many miRNAs are expressed in a tissue- and disease-specific manner and are implicated in a range of biological processes, including those occurring in the liver [18,19]. Cells secrete miRNAs through exosomes and extracellular vesicles into bodily fluids, such as saliva, urine, and blood [20], where they remain stable in relative abundance. This facilitates their detection and quantification for clinical diagnostic applications by the minimally invasive procedure of blood sampling, thereby reducing the need for liver biopsy [19].

A number of miRNAs have been associated with various aspects of liver disease, including hepatitis infection, fibrosis, cirrhosis, and HCC [21]. Among them, five miRNAs—let-7a, miR-21, miR-29a, miR-34a, and miR-122—qualify as candidate blood biomarkers based on their participation and differential regulation in inflammatory processes and tissue injury, including but not limited to the liver, and on their ease of detection in extracellular fluids [22,23,24,25,26,27,28,29,30,31,32,33]. Though involved in a variety of physiological and pathological conditions across multiple organ systems, these miRNAs stood out for their reported relevance to liver pathology, particularly in fibrotic mechanisms. Notably, most supporting studies focused on chronic liver diseases, such as viral hepatitis or non-alcoholic fatty liver disease (NAFLD), where inflammation is a key driver. In contrast, liver fibrosis in β-thalassaemia primarily arises from iron overload, and it remains unclear whether the same miRNAs are equally informative in this distinct context. For comparison, miR-16 and miR-221 may be suitable as endogenous reference miRNAs based on their widespread use as constitutively expressed controls in plasma-based assays (Table 1).

The aim of this study was to evaluate the clinical utility of five selected miRNAs as biomarkers for the detection of liver fibrosis and, moreover, for the discrimination of fibrosis stages, against miR-16 and miR-221 as reference miRNAs. Additionally, a logistic regression analysis was conducted with the objective of assessing the discriminatory power of the candidate miRNAs between the liver fibrosis stages in β-thalassaemia patients.

## 2. Results

### 2.1. Patient Population, Demographics, and Clinical Information

Among the 40 β-thalassaemia patients, liver fibrosis was classified as F0–F1 in 15 patients, stage F2 in 12, stage F3 in 6, and stage F4 in 7. Ages (Mean ± SD) were 48.4 ± 9.2 years (F0–F1), 51.7 ± 7.3 years (F2), 50.5 ± 3.9 years (F3), and 57.7 ± 10.3 years (F4) (Table 2, Figure 1). Importantly, the liver fibrosis groups showed no significant differences in mean age (*p* > 0.4 for all comparisons) based on the Kruskal–Wallis test with Dunn’s multiple comparison test. Mean liver stiffness measurements (LSM ± SD) were 5.84 ± 1.07 kPa (F0–F1), 6.84 ± 0.89 kPa (F2), 10.35 ± 1.27 kPa (F3), and 14.96 ± 3.91 kPa (F4) (Table 2). Additionally, a total of twelve healthy control samples were used to evaluate the dependence of expression for candidate reference miRNAs miR-16 and miR-221, for the β-thalassaemia state, and for age-dependence of expression.

### 2.2. Validation of Endogenous Reference miRNAs, miR-16, and miR-221

The utility of miR-16 and miR-221 as reference miRNAs in β-thalassaemia was confirmed by multiple assessments. First, the age distribution across all patient samples (F0–F1 to F4) and a group of six healthy samples of appropriate age was assessed. No statistically significant differences (*p* = 0.198) were observed, suggesting that age was unlikely to confound the expression of these reference miRNAs in corresponding expression analyses for these samples (Appendix A). Subsequently, the raw Ct values of miR-16 and miR-221 were assessed across the same samples, and no significant variation between groups was observed (*p* = 0.247 and *p* = 0.584, respectively), indicating the absence of differential expression based on fibrosis state and β-thalassaemia disease state and severity. Additionally, the arithmetic mean of Ct values from both miRNAs was evaluated, also showing no significant differences (*p* = 0.379) (Figure 2), supporting the stability of these miRNAs and their combined use as endogenous controls for normalisation. A comparison of six older vs six younger healthy samples with significant groupwise age differences (*p* = 0.001) (Appendix A), for expression of miR-16 (*p* = 0.18), miR-221 (*p* = 0.49), and their average (*p* = 0.18) further confirmed the robustness of expression for both miRNAs for varying ages of subjects (Appendix A).

Finally, validation was carried out through a meta-analysis using the Gene Expression Omnibus (GEO) dataset GSE160879 [58] to compare expression levels of miR-16 (assessed as pre-miRNAs miR-16-1 and miR-16-2) and miR-221 between early-stage (F1–F2) and advanced-stage (F4) fibrosis samples. No statistically significant differences were observed (*p* = 0.50, *p* = 0.69, and *p* = 0.59, respectively) as independent evidence of robustness for the expression of both miRNAs for different severities of fibrosis (Figure 3).

In conclusion, these findings justify the use of miR-16 and miR-221 as stable endogenous reference miRNAs in our patient cohort and, more generally, for comparisons across groups with different fibrotic, age, and β-thalassaemia states.

### 2.3. Expression Levels of the Candidate Plasma miRNAs in β-Thalassaemia Patients with Liver Fibrosis

The expression levels of the candidate miRNAs—let-7a, miR-21, miR-29a, miR-34a, and miR-122—were analysed using RT-qPCR in β-thalassaemia patients with liver fibrosis stages ranging from F0–F1 to F4. Median ΔCt values with interquartile ranges (IQR) for each group are shown in Table 3, and are graphically represented as box plots in Figure 4.

Let-7a expression remained relatively stable, with a slight decrease in F3 and F4 compared to F0–F1 and F2. miR-21 showed a gradual increase from F0–F1 to F2 and F3, followed by a decrease in F4. miR-29a displayed little variation across F0–F1 and F2, with lower levels in F3 and F4. miR-34a gradually decreased from F0–F1 to F2 and F3, followed by a slight increase in stage F4. miR-122 showed the greatest variation, with reduced expression in F2 and a relative increase in F4 compared to F0–F1 (Table 3, Figure 4).

Statistical comparisons across stages showed no significant differences in expression for let-7a (*p* = 0.37), miR-21 (*p* = 0.11), miR-29a (*p* = 0.69), miR-34a (*p* = 0.40), or miR-122 (*p* = 0.44) (Table 3). These findings indicate that the expression levels of the tested miRNAs do not significantly vary with fibrosis progression in this patient cohort.

### 2.4. Diagnostic Accuracy of Candidate miRNAs for Detection of Liver Fibrosis Stages in β-Thalassaemia Patients

Logistic regression was performed to assess the discriminatory ability of individual miRNAs between specified fibrosis stage groupings. Odds Ratios (OR) with 95% confidence intervals (CI) and *p*-values are reported. Receiver operating characteristic (ROC) analysis provided the AUC for each model, and the Youden Index was used to determine the optimal cutoff for classification. Highly tentative analyses indicated that several candidate miRNAs showed fair discriminatory ability in selected pairwise stage comparisons, with AUC values ranging from 0.720 to 0.798; however, none of these models reached statistical significance (*p* > 0.05) (Appendix A, Appendix A). To explore whether panels of miRNAs could improve discriminatory power compared to individual miRNAs, multivariate logistic regression was performed, followed by ROC analysis. Several two- to five-miRNA panels achieved higher AUCs, up to 0.917 in specific pairwise stage comparisons; however, none of these models reached statistical significance (*p* > 0.05) (Appendix A).

## 3. Discussion

In this study, we examined the diagnostic potential of five plasma miRNAs—let-7a, miR-21, miR-29a, miR-34a, and miR-122—as candidate biomarkers for distinguishing between liver fibrosis stages in β-thalassaemia patients. Although miRNAs are increasingly being studied as non-invasive markers for a variety of hepatic conditions, little is known about their potential utility as biomarkers of liver fibrosis in patients with β-thalassaemia.

To facilitate our analyses, we first validated miR-16 and miR-221 as stably expressed reference miRNAs for our cohort, unaffected by fibrotic stage, age, and β-thalassaemia. Their employment for the normalization of our five candidate miRNA biomarkers showed that there were no statistically significant variations in the expression levels of any of the candidate miRNAs across the different stages of fibrosis (Figure 4). However, ROC curve analyses showed that some miRNAs demonstrated moderate diagnostic potential when comparing certain liver fibrosis stages, albeit without statistical significance. For example, miR-21 showed fair discriminatory performance between stages F3 and F4, F0–F1 and F3, and F2 and F3 with AUC values of 0.798, 0.795, and 0.729, respectively. Let-7a could distinguish between stages F2 and F4, and F2 and F3 with AUC scores of 0.750 and 0.743, respectively. miR-34a showed better differential performance than let-7a with an AUC score of 0.767 between stages F3 and F4 (Appendix A, Appendix A). Despite its reported low baseline abundance in plasma (Table 1) [57], miR-29a was quantifiable with moderate Ct values in our study, albeit with only fair diagnostic performance across comparisons between F3 and F4, with an AUC of 0.733. On this note, the overall promising AUC values across miRNAs must be interpreted with caution, as they are based on a relatively small sample size, in particular for the F3 and F4 groups (Appendix A), which limits statistical power and may have prevented the detection of subtle, yet biologically meaningful, differences. Even though groupwise differences were not significant, we included ROC analyses as indicative of potential discriminatory trends for follow-up by higher-powered biomarker discovery studies. In this study, the ROC analyses should therefore be regarded as preliminary and hypothesis-generating only, and validation in larger, independent cohorts is essential to establish their true discriminatory value.

Identifying miRNAs that vary with liver fibrosis stage would have important clinical implications for β-thalassaemia, where iron overload drives disease progression [2,59]. Biomarkers capable of distinguishing fibrosis severity could improve monitoring, guide chelation therapy decisions, and potentially offer early warnings of advanced disease [60]. However, none of the five candidate miRNAs showed significant differences in expression across liver fibrosis stages. Although their expression appeared relatively consistent in our cohort across samples, they were not treated as endogenous controls, because they had been reported as differentially expressed for liver fibrosis. Moreover, expression values for the five candidate miRNAs are shown throughout after normalization to miR-16 and miR-221, which are well-established as stable reference miRNAs in plasma (Table 1) and showed consistent expression in our dataset (Figure 2 and Appendix A) and in third-party fibrosis data used for meta-analysis (Figure 3). Importantly, to our knowledge, this is the first study to establish miR-16 and miR-221 as suitable endogenous reference miRNAs in β-thalassaemia patients with liver fibrosis, and it does so both based on our own data and based on meta-analysis of third-party RNA-seq data. Conversely, the apparent stability of miR-16/miR-221-normalized expression for the candidate miRNAs in this study across fibrotic stages in thalassemia should be regarded as incidental. Nevertheless, this observation raises the question of why the discriminative power of let-7a, miR-21, miR-29a, miR-34a, and miR-122 for fibrosis stages in this patient cohort was limited, given their generally faithful reflection of liver injury and fibrotic changes in other contexts [27,30,31,61]. More specifically, all five miRNAs were previously linked to liver-related pathology and fibrotic mechanisms such as inflammation, fibrogenesis, and hepatocellular injury [21,25,62,63]. Still, these miRNAs, particularly miR-21 and let-7a, are involved in multiple disease pathways, including cancers, cardiovascular diseases, and neurological disorders, as shown in the Human MicroRNA Disease Database (HMDD v4.0) [64]. Although this lack of specificity may limit their stand-alone use, several studies have reported strong performance of these same miRNAs in fibrosis settings driven by viral hepatitis or NAFLD, often with AUCs exceeding 0.800 [65,66,67,68] while corresponding data in β-thalassaemia remain scarce. It is worth considering that in contrast to viral hepatitis and NAFLD, the fibrotic context of β-thalassaemia is dominated by iron toxicity rather than inflammation, which may explain the lower AUC values and lack of significant differences observed in our study and further supports the idea that disease-specific biomarker validation is essential. These considerations and our results suggest that a wider assessment of suitable miRNAs, beyond those identified for liver fibrosis in other disease contexts, may help identify informative miRNAs as circulating biomarkers for fibrosis staging in β-thalassaemia.

Taken together, this study provides a useful baseline for detecting miRNA biomarkers in β-thalassaemia-related liver disease. In a field with limited prior research, it offers a standardized methodology and statistically evaluable data from a genetically homogenous patient group [69], using widely used liver fibrosis staging methods as a reference. Beyond sample size, several limiting factors impacted our results. These include operator-dependent variability in elastography-based fibrosis staging across three clinics. In addition, while TE is widely applied in clinical hepatology, its validation and interpretation in β-thalassaemia needs to consider the wider disease pathology. For example, Fraquelli et al. found a significant correlation between TE-measured fibrosis stiffness and fibrosis stage (r = 0.73, *p* = 0.003), but noted that iron overload may influence TE results [70]. In our cohort, both TE and SWE were variably applied across clinics, which may have introduced method-related variability and further highlights the need for validation of elastography-based staging in β-thalassaemia.

Beyond imaging, another limitation is the biomarker source itself. β-thalassaemia significantly alters the circulating miRNA landscape, potentially confounding liver-derived signatures. For example, both let-7a-5p and miR-122-5p have been reported as deregulated in thalassaemia patients [70,71,72]. Such widespread changes in circulating miRNA expression in β-thalassaemia thus lower the sensitivity for the detection of liver-specific miRNA signals. In addition, other factors common in β-thalassaemia—such as iron overload, comorbidities, and general physiological differences—can also affect circulating miRNA levels. These combined effects likely reduce the sensitivity of plasma miRNAs for detecting liver fibrosis in this patient group. In addition to single-marker analyses, we also explored whether combining miRNAs in multivariate models could improve discriminatory power. Several two- to five-miRNA panels achieved higher AUC values than any of the individual miRNAs, albeit without reaching statistical significance and with wide confidence intervals as a consequence of the limited cohort size (Appendix A). Our multivariate analyses, therefore, highlight the potential of multi-marker panels, but are exploratory and indicate the necessity of validation in larger, independent cohorts.

Moving forward, future studies should consider larger, multi-centre cohorts to increase power and control for possible clinic-to-clinic variation. In parallel, a wide range of hepatic miRNAs and specifically the use of miRNA panels—rather than single markers—should be explored to capture broader disease signatures and improve discriminatory accuracy. Additionally, longitudinal research would reveal the influence of time and treatment on levels of miRNAs and their utility as biomarkers. To move toward validation and integration of miRNA biomarker technology into routine diagnostics, the generalizability of findings needs to be established. This could be achieved by applying the approach to additional patient populations and by comparing or combining miRNA-based assessments with quantitative iron overload measures, such as T2* MRI, as well as with other potential liver fibrosis markers and source materials. In this context, it is important to note that a variety of sources of miRNAs are currently being explored [73,74,75] and that in animal models, detection of liver-related miRNA in urine has been correlated with that in blood [76]. Methodology and sensitivity of urinary miRNA detection are an active field of research, including for hepatic miRNA markers [77,78,79,80,81]. This development suggests that a comparison of the performance of urinary to circulating miRNAs for β-thalassaemia patients may be worthwhile, with the potential to pave the way towards a non-invasive urinary miRNA assay that would help minimize the interference of any thalassaemia-related changes in the detection of liver-specific miRNA biomarkers.

In conclusion, further research is needed before minimally invasive miRNA expression analysis can become part of routine patient management for liver fibrosis in β-thalassaemia, even as additional miRNA source material may be explored. While individual miRNAs such as let-7a, miR-21, miR-34a, and miR-122 show some potential as fibrosis-related biomarkers, their use in clinical decision-making is likely to depend on their integration into broader multi-marker strategies. This work lays the groundwork for such efforts and highlights key challenges in applying miRNA diagnostics to complex multisystemic disorders like β-thalassaemia.

## 4. Materials and Methods

### 4.1. Study Population

A total of 40 patients with β-thalassaemia were enrolled in this study as part of their routine medical regimen for the detection and evaluation of liver fibrosis. Liver stiffness was assessed by elastography, either using TE (FibroScan^®^, Echosens, Paris, France), SWE on Philips Epiq 7 and Philips Affiniti 70G (Philips Healthcare, Andover, MA, USA), or GE Logiq series systems (models E9, E10, and S8; GE Healthcare, Chicago, IL, USA), depending on the diagnostic centre. Blood samples from study subjects were collected from three separate hospital locations in Cyprus between March 2021 and February 2022. The patients were categorized based on their stage of liver fibrosis (F0–F1, F2, F3, and F4), as determined by elastography results and in line with the widely used METAVIR scoring system, as follows: F0–F1, no fibrosis/portal fibrosis without septa; F2, portal fibrosis with few septa; F3, bridging septa between central and portal veins; and F4, cirrhosis (Table 2) [82].

Some patients were classified as stage F0, some as F1, and others as F0–F1, due to differences in reporting practices among the participating clinics. To ensure consistency in grouping and to facilitate statistical analysis, patients with F0, F1, and F0–F1 results were grouped together under a unified “F0–F1” category, reflecting their status as absent/minimal to early-stage fibrosis. In addition to thalassaemic samples, a total of twelve healthy control samples were used to evaluate invariability of expression for candidate reference miRNAs miR-16 and miR-221, (a) for β-thalassaemia state by inclusion of age-similar healthy samples aged 44.50 ± 5.85 (*n* = 6) in comparison with β-thalassaemia samples, and (b) for age-dependence of expression in comparisons for older healthy individuals with significantly younger individuals aged 28.33 ± 1.97 (*n* = 6; *p* = 0.001) (Appendix A, Appendix A).

This study was approved by the Cypriot National Bioethics Committee, and written informed consent was obtained from each participant.

### 4.2. Blood Sampling

Peripheral blood samples were collected from each individual directly into BD Vacutainer^®^ Venous Blood Collection Tubes containing EDTA as anticoagulant (BD, Plymouth, UK). The blood samples were stored at room temperature until plasma extraction was carried out, which was initiated within 4 h of collection. The samples were mixed by inverting the blood collection tubes from 8 to 10 times, after which they were centrifuged for 20 min at 1300× *g* at 4 °C. The upper plasma phase was then transferred into a new collection tube and centrifuged again for 10 min at 16,000× *g* at 4 °C. The plasma samples were divided into separate aliquots, ensuring that the debris pellet was not disturbed, and stored at −80 °C until RNA isolation was performed.

### 4.3. RNA Isolation

The plasma samples were thawed and centrifuged for 5 min at 16,000× *g* at 4 °C to remove any residual cellular debris. Subsequently, the supernatant was transferred into a new collection tube, after which RNA isolation was performed. Total RNA, including small RNAs, was isolated from 300 μL of plasma using the NucleoSpin^®^ miRNA Plasma kit (Macherey-Nagel, Düren, Germany) according to the manufacturer’s instructions. Small RNAs were quantified using the Qubit microRNA Assay Kit and the Qubit Flex Fluorometer (Thermo Fisher Scientific, Waltham, MA, USA).

### 4.4. Reverse Transcription

The TaqMan microRNA Reverse Transcription Kit was used with the TaqMan microRNA RT primers (Life Technologies, Carlsbad, CA, USA) for the following miRNAs: hsa-miR-16-5p, hsa-miR-21-5p, has-miR-29a-5p, hsa-miR-34a-5p, hsa-miR-122-5p, hsa-miR-221-3p, and hsa-let-7a-5p (Table 1). The reverse transcription reactions were carried out according to the manufacturer’s instructions and using a Veriti™ Thermal Cycler (Applied Biosystems, Foster City, CA, USA) at 16 °C for 30 min, 42 °C for 30 min, and 85 °C for 5 min. The resulting cDNA was stored at −20 °C until further analysis.

### 4.5. Reverse-Transcription Quantitative Polymerase Chain Reaction (RT-qPCR)

Triplicate RT-qPCR reactions were performed according to the manufacturer’s instructions using the 20× TaqMan Small RNA Assay buffer (Life Technologies, USA) and the Universal PCR Master Mix (Life Technologies, USA). The RT-qPCR reactions were carried out using the 7900HT Fast Real-Time PCR System (Applied Biosystems, USA), and results were analysed using the Sequence Detection System (SDS) Software version 2.4 (Applied Biosystems, USA).

### 4.6. Statistical Analysis

Statistical tests were performed and figures were generated using GraphPad Prism version 9 (GraphPad Software Inc., La Jolla, CA, USA) and R software (version 4.5.1; R Foundation for Statistical Computing, Vienna, Austria).

For groupwise comparisons, data normality was assessed using the Shapiro–Wilk test, and homogeneity of variances was evaluated using Levene’s test prior to one-way ANOVA. One-way ANOVA with Tukey’s post-hoc test was applied when both assumptions were met, while Kruskal–Wallis with Dunn’s multiple comparisons post-hoc test was used otherwise. Outliers in miRNA expression were identified using studentised residuals from linear models of ΔCt values across fibrosis stages. Samples with absolute residuals greater than 2 in more than 10% of the miRNAs were excluded from further analysis.

Expression levels (ΔCt) for each candidate miRNA were calculated by normalising the average Ct values of each candidate miRNA to the arithmetic mean of miR-16 and miR-221, which were used as the endogenous controls.

To validate the stability of these endogenous controls, expression data for miR-16-1, miR-16-2, and miR-221 were retrieved from the GSE160879 dataset [58]. Normality was assessed using the Shapiro–Wilk test, and comparisons between groups were conducted using Welch’s *t*-test or the Wilcoxon rank-sum test, depending on distribution. Fold changes were calculated as the ratio of mean expression in F4 to F1–F2, and Cohen’s d was computed to evaluate effect size.

Logistic regression was performed to assess the discriminatory ability of candidate miRNAs between specified fibrosis stage groupings using both univariate and multivariate models, the latter for combinations of between two and five miRNAs. Odds ratios (ORs) with 95% confidence intervals and *p*-values are reported. ROC analysis provided the AUC for each model, and the Youden Index was used to determine the optimal cutoff for classification based on the maximum sum of sensitivity and (specificity—1). ROC curves were generated using the pROC package in R.

All statistical tests were two-sided, and a threshold of 0.05 was applied for all groupwise comparisons.

## Figures and Tables

**Figure 1 ijms-26-09543-f001:**
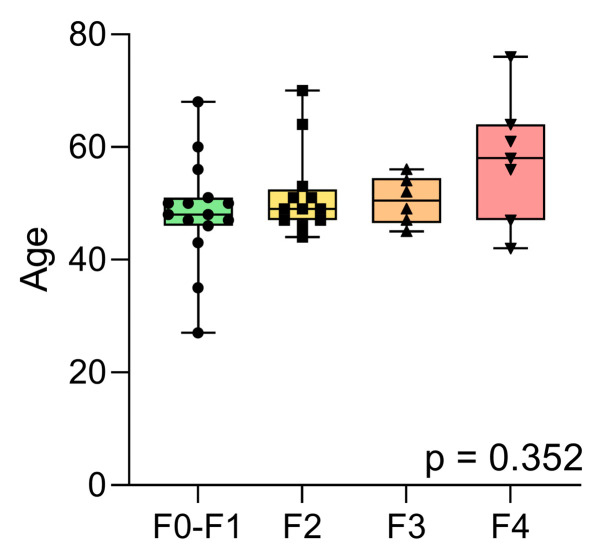
Patient ages by fibrosis stage. Distribution of ages among β-thalassaemia patients with various stages of liver fibrosis (F0–F1, F2, F3, and F4). Each point represents an individual participant’s age, and boxplots indicate the median, interquartile range, and the range within each group. The *p*-value indicated is that of the groupwise age comparison performed using the Kruskal–Wallis test.

**Figure 2 ijms-26-09543-f002:**
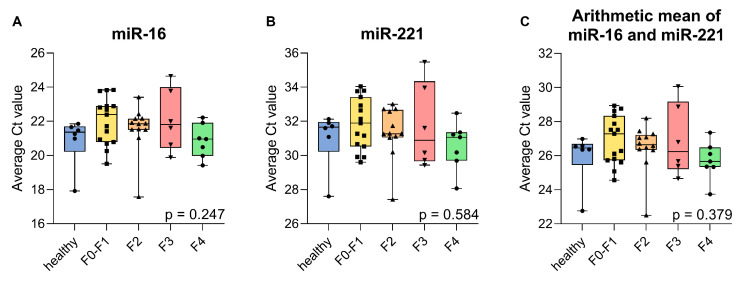
Groupwise comparison of average Ct values for (**A**) miR-16, (**B**) miR-221, and (**C**) arithmetic mean of miR-16 and miR-221 across liver fibrosis stages and healthy controls. Ct values represent the arithmetic mean of triplicate RT-qPCR measurements. Group sizes were healthy (*n* = 6), F0–F1 (*n* = 15), F2 (*n* = 12), F3 (*n* = 7), and F4 (*n* = 6). Each boxplot shows the median, interquartile range, and range of the average Ct values per sample; each point represents an individual sample. Statistical differences among groups were assessed using the Kruskal–Wallis test. Corresponding *p*-values are indicated on the lower right-hand side of each panel.

**Figure 3 ijms-26-09543-f003:**
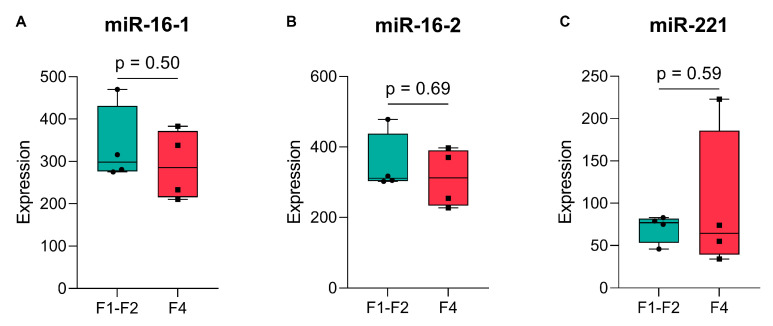
Meta-analysis of expression levels of (**A**) miR-16-1, (**B**) miR-16-2, and (**C**) miR-221 in early-stage (F1–F2) versus advanced-stage (F4) liver fibrosis. Boxplots display pooled liver tissue expression data extracted from the GEO dataset GSE160879 [58]. miR-16-1 and miR-16-2 represent distinct loci that give rise to the same mature miR-16-5p sequence. Each dot represents an individual sample. Boxplots indicate the median, interquartile range, and full range within each group. Statistical comparisons between fibrosis stages were performed for each miRNA: Welch’s *t*-test for miR-16-1 and miR-221, and Wilcoxon rank-sum test for miR-16-2. Corresponding *p*-values are indicated.

**Figure 4 ijms-26-09543-f004:**
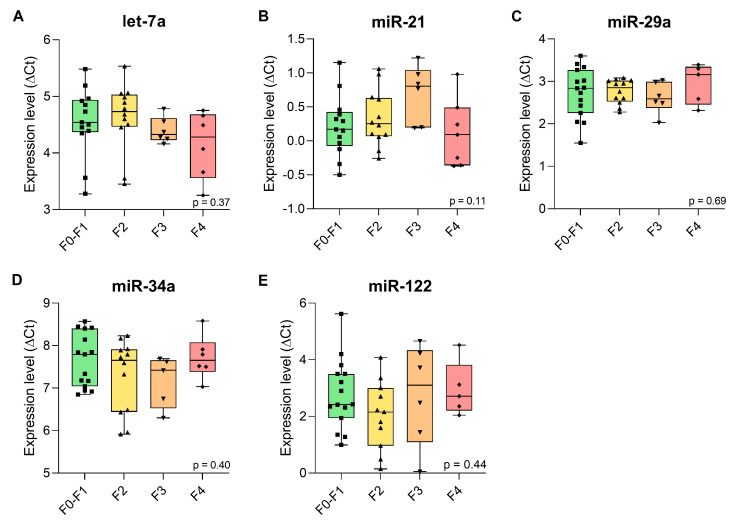
Expression levels (ΔCt) of candidate miRNAs in plasma samples from β-thalassaemia patients across liver fibrosis stages (F0–F1, F2, F3, and F4). Boxplots represent the distribution of ΔCt values for (**A**) let-7a, (**B**) miR-21, (**C**) miR-29a, (**D**) miR-34a, (**E**) miR-122. Each point represents an individual participant’s miRNA expression level. Boxplots indicate the median, interquartile range (IQR), and total range within each group.

**Table 1 ijms-26-09543-t001:** Candidate plasma miRNAs with sequences, inclusion rationale, and reported abundance.

miRNA	miRNA Sequence (5′ to 3′)	Reason for Inclusion (Reported Abundance *)	References
hsa-miR-16-5p	UAGCAGCACGUAAAUAUUGGCG	Reported as endogenous reference miRNA in plasma-based assays (16.7 [15.4, 17.9])	[34,35,36,37,38]
hsa-miR-221-3p	UGAGGUAGUAGGUUGUAUAGUU	Reported as endogenous reference miRNA in plasma-based assays (5.9 [3.9, 6.9])	[39,40,41]
hsa-let-7a-5p	UAGCUUAUCAGACUGAUGUUGA	Downregulated circulating hepatic biomarker (7.25 [5.91, 8.39])	[42,43,44,45]
hsa-miR-21-5p	ACUGAUUUCUUUUGGUGUUCAG	Upregulated circulating hepatic biomarker (9.4 [8.5, 10.4])	[37,38,46,47,48,49,50]
hsa-miR-29a-5p	UGGCAGUGUCUUAGCUGGUUGU	Upregulated circulating hepatic biomarker (0 [0, 0.67])	[51,52]
hsa-miR-34a-5p	UGGAGUGUGACAAUGGUGUUUG	Upregulated circulating hepatic biomarker (2.4 [0, 3.6])	[37,53,54,55]
hsa-miR-122-5p	AGCUACAUUGUCUGCUGGGUUUC	Upregulated circulating hepatic biomarker (10.1 [8.1, 12.0])	[38,55,56]

* Abundance numbers are median and interquartile range (IQR) as reported in Jehn et al. (2024) [57].

**Table 2 ijms-26-09543-t002:** Demographic and liver stiffness measurement (LSM) data for the 40 β-thalassaemia patients enrolled in this study.

Fibrosis Stage	F0–F1	F2	F3	F4
Age: mean ± SD	48.4 ± 9.2	51.7 ± 7.3	50.5 ± 3.9	57.7 ± 10.3
Number of patients, *n*	15	12	6	7
Patients				
Male	5 (33.3%)	8 (66.7%)	4 (66.7%)	4 (57.1%)
Female	10 (66.6%)	4 (33.3%)	2 (33.3%)	3 (42.9%)
LSM (kPa): mean ± SD	5.84 ± 1.07	6.84 ± 0.89	10.35 ± 1.27	14.96 ± 3.91

**Table 3 ijms-26-09543-t003:** Median ΔCt values (IQR) of candidate miRNAs across fibrosis stages, with groupwise comparisons *.

Fibrosis Stage (*n* **)	let-7a	miR-21	miR-29a	miR-34a	miR-122
F0–F1 (15)	4.52 (4.15–4.93)	0.17 (−0.075–0.43)	2.84 (2.26–3.27)	7.79 (7.04–8.40)	2.42 (1.94–3.50)
F2 (12)	4.73 (4.46–5.03)	0.25 (0.068–0.63)	2.85 (2.53–3.02)	7.66 (6.44–7.91)	2.15 (0.96–3.00)
F3 (6)	4.33 (4.23–4.62)	0.81 (0.20–1.04)	2.59 (2.38–2.99)	7.42 (6.53–7.66)	3.10 (1.09–4.34)
F4 (7)	4.49 (3.66–4.75)	0.09 (−0.36–0.49)	3.23 (2.52–4.33)	7.66 (7.38–8.08)	2.71 (2.20–3.82)
*p*-value	0.37	0.11	0.69	0.40	0.44

* Data normality was assessed using the Shapiro–Wilk test and homogeneity of variances using Levene’s test. Both assumptions were met for let-7a, miR-21, miR-29a, and miR-122 (*p* > 0.05), permitting the use of one-way ANOVA with Tukey’s post-hoc test, whereas miR-34a did not meet the normality assumption and was analysed with Kruskal–Wallis test with Dunn’s multiple comparisons post-hoc test. The *p*-values shown are those for one-way ANOVA/Kruskal–Wallis tests. ** Original number of patients per fibrosis stage. Actual *n* used to calculate median ΔCt and interquartile ranges (IQR) varies per miRNA due to outlier exclusion (see Section 4.6. Statistical Analysis and Appendix A).

## Data Availability

Data is contained within the article and Appendix A.

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
