# Peer review of "Evaluation of Five Plasma miRNAs as Biomarkers for Minimally Invasive Staging of Liver Fibrosis in β-Thalassaemia Patients"

_ijms, 2025, doi:10.3390/ijms26199543_

Round 1

Reviewer 1 Report

Comments and Suggestions for Authors

The authors wanted to test an interesting hypothesis. However, the quality and quantity of the material are insufficient for such a test. In individual groups, the number of patients is 2, 3, 4 people (if divided by the degree of fibrosis and gender). And even without division by gender, with stage 3 and 4 fibrosis, there are 6 and 7 people, respectively. What kind of logistic regression analysis is this? Or determining the sensitivity of specificity, ROC analysis? With such material, such approaches look at least unfounded or a purely theoretical attempt, on which no reasonable conclusions can be made. And the presentation of the results as mean values ​​with a standard deviation is also puzzling. The groups are very small, so there is no reason to expect a normal distribution of indicators. Accordingly, the results should be presented as a median of 25 and 75 quartiles.
"The blood samples were stored at room temperature until plasma extraction was carried out, which was initiated within 4 hours of collection." Question: how did such a long storage at room temperature affect the amount of RNA? It cannot be ruled out that if the plasma samples had been placed in a freezer at -80 as quickly as possible, the results could have been different.

Author Response

[Line numbers given below refer to numbering in the v2 manuscript.]

[Comment 1] The authors wanted to test an interesting hypothesis. However, the quality and quantity of the material are insufficient for such a test. In individual groups, the number of patients is 2, 3, 4 people (if divided by the degree of fibrosis and gender). And even without division by gender, with stage 3 and 4 fibrosis, there are 6 and 7 people, respectively. What kind of logistic regression analysis is this? Or determining the sensitivity of specificity, ROC analysis? With such material, such approaches look at least unfounded or a purely theoretical attempt, on which no reasonable conclusions can be made.

[Response 1] We agree that the small sample size, particularly in stages F3 and F4, limits the robustness of logistic regression and ROC analyses. This is mentioned in Discussion (lines 251-255) “On this note, the overall promising AUC values across miRNAs must be interpreted with caution, as they are based on a relatively small sample size, in particular for the F3 and F4 groups (Table S1), which limits statistical power and may have prevented the detection of subtle, yet biologically meaningful, differences.” Our analyses were simple univariate logistic regression models, applied separately to each candidate miRNA. AUC values, Youden index values, sensitivity, specificity and ROC curves were generated for exploratory purposes only. To reflect this limitation more clearly, we have moved Table 4 and Figure 5 from the main manuscript to the Supplementary data (now Table S3 and Figure S4). In addition, the Results section (2.4) (lines 221-225) was revised to state “Highly tentative analyses indicated that several candidate miRNAs showed fair discriminatory ability in selected pairwise stage comparisons, with AUC values ranging from 0.720 to 0.798; however, none of these models reached statistical significance (p > 0.05) (Table S3, Figure S4).” Furthermore, in Discussion (lines 257-259), we added the following clarification: “In this study, the ROC analyses should therefore be regarded as preliminary and hypothesis-regarding only, and validation in larger, independent cohorts is essential to establish their true discriminatory value.

[Comment 2] And the presentation of the results as mean values with a standard deviation is also puzzling. The groups are very small, so there is no reason to expect a normal distribution of indicators. Accordingly, the results should be presented as a median of 25 and 75 quartiles.

[Response 2] As described in the Methods, we assessed the normality (Shapiro-Wilk test) and homogeneity of variances (Levene’s test) prior to selecting parametric (ANOVA with Tukey’s post-hoc) or non-parametric (Kruskal-Wallis with Dunn’s post-hoc) analyses to confirm nominal compliance with the tests used. Although these analyses in their sample size are equivalent to and in their transparency as to the tests employed go beyond many other studies that employ mean ± SD (or, wrongly, mean ± SEM) as descriptive statistics (e.g. PMID 38139055, 35867132, 32683713, 38123450), we concede that bioscience publications often fall short in their statistical rigor and should not serve as our lodestar; we agree that across all groups with at times small group sizes, median and interquartile range (IQR) represent more robust descriptive statistics measures. While the box plots in Figure 4 for our original submission already displayed median and 25th / 75th percentiles, we have now revised Table 3 to report Median ΔCt values with IQR, while keeping the inferential statistics unchanged. In addition, the Results section (2.3) has been revised to remove detailed numerical reporting and instead describe overall expression trends in line with the new format.

[Comment 3] "The blood samples were stored at room temperature until plasma extraction was carried out, which was initiated within 4 hours of collection." Question: how did such a long storage at room temperature affect the amount of RNA? It cannot be ruled out that if the plasma samples had been placed in a freezer at -80 as quickly as possible, the results could have been different.

[Response 3] In our study, we were alerted by the clinicians immediately upon blood collection from the patients, and typically retrieved the blood samples within 1-2 hours. On rare occasions, sample arrival in our laboratory from the most distant clinic was delayed by a maximum of 4 hours at room temperature before processing (hence the phrase “within 4 hours” in the M&M section). After sample receipt in our laboratory, plasma extraction was carried out without delay, and plasma aliquots were stored at -80 °C immediately.

Of note, our handling was very conservative compared to general practice. Stability of circulating miRNAs under routine collection and processing conditions, including temporary storage at room temperature, is part of their appeal as biomarkers for routine clinical use. Previous studies showed that miRNAs in whole blood are stable at room temperature for 12–24 hours (Benson and Skaar 2013, PMID: 23886700; Glinge et al. 2017, PMID: 28151938). Based on these third-party reports and our processing protocol, we believe the integrity of our samples was preserved.

Reviewer 2 Report

Comments and Suggestions for Authors

Authors present a paper suggesting the potential role of five plasma miRNAs as non-invasive biomarkers of liver fibrosis in patients with iron-overload due to beta-thalasemia. Some details require attention

Minor

Introduction

Line 93, “surgery for” should be deleted, since surgery is not mandatory to retrieve liver biopsy, it can be obtained by percutaneous or trans-jugular procedures.

Methods, it would be of interest to include the type of elastography that was applied to assess liver fibrosis, as well as the description of the FibroScan equipment used.

Major

Authors describe miRNAS that show a steady expression during liver fibrosis (miR-16-1, miR-16-2 and miR-221) thus can be used as endogenous reference. Analysis of their target miRNAS (Let-7a, miR-21, miR-29a, miR-34a, and miR-122) also show a steady expression in the different fibrosis stages, no significant differences were observed. A brief explanation should be added to the discussion regarding why the first ones are considered endogenous controls whereas the latter are not, even though both groups of miRNAS behave in a similar fashion.

It is interesting to note that despite not showing significant differences in expression among the fibrosis stages, authors still found values of AUROC higher than 0.7.  Discussion is needed supporting the rationale of using non-significant data to assess sensitivity, specificity and ROC curves, and the cautions that should be taken regarding these data.

Authors analyzed every miRNA separately, however, in biomarker development it is widely used the multiplexing approach, where a combination of analytes improves specificity, sensitivity and AUROC. The combination of miRNAs might be worth testing, for example miR-21+miR-34a for F0-F1 vs F3.

Discussion should also include the limitations of the study, including FibroScan validation of for this etiology, the modality of elastography applied, the fact that iron overload has been associated with poor performance of elastography (at least with magnetic resonance elastography).

Author Response

[Line numbers given below refer to numbering in the v2 manuscript.]

Authors present a paper suggesting the potential role of five plasma miRNAs as non-invasive biomarkers of liver fibrosis in patients with iron-overload due to beta-thalasemia. Some details require attention

Minor

[Comment 1] Introduction Line 93, “surgery for” should be deleted, since surgery is not mandatory to retrieve liver biopsy, it can be obtained by percutaneous or trans-jugular procedures.

[Response 1] We thank the reviewer for this observation. As suggested, we have removed the phrase “surgery for” from the Introduction. The revised sentence (lines 94-95), now reads “…for clinical diagnostic applications by the minimally invasive procedure of blood sampling, thereby reducing the need for liver biopsy”.

[Comment 2] Methods, it would be of interest to include the type of elastography that was applied to assess liver fibrosis, as well as the description of the FibroScan equipment used.

[Response 2] We have revised the Introduction to briefly describe both, transient elastography and shear wave elastography, as widely used imaging modalities (lines 66-72): “Instead, non-invasive biochemical and imaging techniques are used for regular evaluation of the liver [8], such as ultrasound-based elastography. The most established method is transient elastography (TE), which assesses liver stiffness as an indicative marker of fibrosis. Additionally, shear wave elastography (SWE) has been introduced, enabling real-time two-dimensional assessment of liver stiffness [9]. Both techniques are widely used in clinical practice; however, each has limitations.

In the Methods, we specified that patients underwent elastography-based staging using either TE or SWE, depending on the clinic (lines 353-357): “Liver stiffness was assessed by elastography, either using TE (FibroScan®, Echosens, Paris, France), SWE on Philips Epiq 7 and Philips Affiniti 70G (Philips Healthcare, Andover, MA, USA) or GE Logiq series systems (models E9, E10, and S8; GE Healthcare, Chicago, IL, USA), depending on the diagnostic center.

Major

[Comment 3] Authors describe miRNAS that show a steady expression during liver fibrosis (miR-16-1, miR-16-2 and miR-221) thus can be used as endogenous reference. Analysis of their target miRNAS (Let-7a, miR-21, miR-29a, miR-34a, and miR-122) also show a steady expression in the different fibrosis stages, no significant differences were observed. A brief explanation should be added to the discussion regarding why the first ones are considered endogenous controls whereas the latter are not, even though both groups of miRNAS behave in a similar fashion.

[Response 3] We have revised the Discussion to clarify the distinction between endogenous reference miRNAs and candidate miRNAs. Accordingly, the following text was added to the Discussion (lines 265-273): “Although their expression appeared relatively consistent in our cohort across samples, they were not treated as endogenous controls, because they had been reported as differentially expressed for liver fibrosis. Moreover, expression values for the candidate miRNAs is consistently shown after normalization to miR-16 and miR-221, which are well-established as stable reference miRNAs in plasma (Table 1) and showed consistent expression in our dataset (Figure 2, Figure S3) and in third-party fibrosis data used for meta-analysis (Figure 3). Importantly, to our knowledge, this is the first study to establish miR-16 and miR-221 as suitable endogenous reference miRNAs in β-thalassaemia patients with liver fibrosis, and it does so both, based on our own data and based on meta-analysis of third-party RNA-seq data. Conversely, the apparent stability of miR-16/miR-221-normalized expression for the candidate miRNAs in this study across fibrotic stages in thalassemia should be regarded as incidental.

[Comment 4] It is interesting to note that despite not showing significant differences in expression among the fibrosis stages, authors still found values of AUROC higher than 0.7.  Discussion is needed supporting the rationale of using non-significant data to assess sensitivity, specificity and ROC curves, and the cautions that should be taken regarding these data.

[Response 4] We agree with this comment, and mindful of the risk of over-interpretation of our data we had therefore emphasized in our Discussion that the ROC results are preliminary and must be interpreted with caution given the small sample size (lines 251-255 and lines 257-259). For further clarification, we added the following text to Discussion (lines 255-257): “Even though groupwise differences were not significant, we included ROC analyses as indicative of potential discriminatory trends for follow-up by higher-powered biomarker discovery studies.

[Comment 5] Authors analyzed every miRNA separately; however, in biomarker development it is widely used the multiplexing approach, where a combination of analytes improves specificity, sensitivity and AUROC. The combination of miRNAs might be worth testing, for example miR-21+miR-34a for F0-F1 vs F3.

[Response 5] We thank the reviewer for this suggestion. To address it, we performed exploratory multivariable logistic regression analyses for all combinations of candidate miRNAs, followed by ROC analysis. The results are provided in Supplementary Tables S4 and S5, and described in the revised manuscript (Results section 2.4, lines 225–230):

To explore whether panels of miRNAs could improve discriminatory power compared to individual miRNAs, multivariate logistic regression was performed followed by ROC analysis. Several two- to five-miRNA panels achieved higher AUCs, up to 0.917 in specific pairwise stage comparisons; however, none of these models reached statistical significance (p > 0.05) (Table S4, Table S5).

In the Discussion (lines 317-323), we added:

In addition to single-marker analyses, we also explored whether combining miRNAs in multivariate models could improve discriminatory power. Several two- to five-miRNA panels achieved higher AUC values than any of the individual miRNAs, albeit as a consequence of the limited cohort size without reaching statistical significance and with wide confidence intervals (Table S4, Table S5). Our multivariate analyses therefore highlight the potential of multi-marker panels, but are exploratory and indicate the necessity of validation in larger, independent cohorts.

Finally, the Materials and Methods (lines 431-433) were updated to clarify:

Logistic regression was performed to assess the discriminatory ability of candidate miRNAs between specified fibrosis stage groupings using both univariate and multivariate models, the latter for combinations of between two and five miRNAs.

[Comment 6] Discussion should also include the limitations of the study, including FibroScan validation of for this etiology, the modality of elastography applied, the fact that iron overload has been associated with poor performance of elastography (at least with magnetic resonance elastography).

[Response 6] We have revised the Discussion accordingly and added the following text (lines 301-308): “In addition, while TE is widely applied in clinical hepatology, its validation and interpretation in β-thalassaemia needs to consider the wider disease pathology. For example, Fraquelli et al. found significant correlation between TE-measured fibrosis stiffness and fibrosis stage (r = 0.73, p = 0.003), but noted that iron overload may influence TE results [70]. In our cohort, both TE and SWE were variably applied across clinics, which may have introduced method-related variability, and further highlights the need for validation of elastography-based liver fibrosis staging in β-thalassaemia.

Round 2

Reviewer 1 Report

Comments and Suggestions for Authors

Accept in present form